# COMPLETE VERIFICATION VIA MULTI-NEURON RELAXATION GUIDED BRANCH-AND-BOUND

**Claudio Ferrari, Mark Niklas Müller, Nikola Jovanović, Martin Vechev**
Department of Computer Science, ETH Zurich, Switzerland
{claudio.ferrari, mark.mueller, nikola.jovanovic, martin.vechev}@inf.ethz.ch

## ABSTRACT

State-of-the-art neural network verifiers are fundamentally based on one of two paradigms: either encoding the whole verification problem via tight multi-neuron convex relaxations or applying a Branch-and-Bound (BaB) procedure leveraging imprecise but fast bounding methods on a large number of easier subproblems. The former can capture complex multi-neuron dependencies but sacrifices completeness due to the inherent limitations of convex relaxations. The latter enables complete verification but becomes increasingly ineffective on larger and more challenging networks. In this work, we present a novel complete verifier which combines the strengths of both paradigms: it leverages multi-neuron relaxations to drastically reduce the number of subproblems generated during the BaB process and an efficient GPU-based dual optimizer to solve the remaining ones. An extensive evaluation demonstrates that our verifier achieves a new state-of-the-art on both established benchmarks as well as networks with significantly higher accuracy than previously considered. The latter result (up to 28% certification gains) indicates meaningful progress towards creating verifiers that can handle practically relevant networks.

## 1 INTRODUCTION

Recent years have witnessed substantial interest in methods for certifying properties of neural networks, ranging from stochastic approaches (Cohen et al., 2019) which construct a robust model from an underlying base classifier to deterministic ones (Gehr et al., 2018; Katz et al., 2017; Xu et al., 2020) that analyze a given network as is (the focus of our work).

**Key Challenge: Scalable and Precise Non-Linearity Handling** Deterministic verification methods can be categorized as complete or incomplete. Recent incomplete verification methods based on propagating and refining a single convex region (Müller et al., 2022; Dathathri et al., 2020; Tjandraatmadja et al., 2020) are limited in precision due to fundamental constraints imposed by convex relaxations. Traditional complete verification approaches based on SMT solvers (Ehlers, 2017) or a single mixed-integer linear programming encoding of a property (Tjeng et al., 2019; Katz et al., 2017) suffer from worst-case exponential complexity and are often unable to compute sound bounds in reasonable time-frames. To address this issue, a Branch-and-Bound approach (Bunel et al., 2020) has been popularized recently: anytime-valid bounds are computed by recursively splitting the problem domain into easier subproblems and deriving bounds on each of these via cheap and less precise methods (Xu et al., 2021; Wang et al., 2021; Palma et al., 2021; Henriksen & Lomuscio, 2021). This approach has proven effective on (smaller) networks where there are relatively few unstable activations and splitting a problem simplifies it substantially. However, for larger networks or those not regularized to be amenable to certification this strategy becomes increasingly ineffective as the larger number of unstable activations makes individual splits less effective, which is exacerbated by the relatively loose underlying bounding methods.

**This Work: Branch-and-Bound guided by Multi-Neuron Constraints** In this work, we propose a novel certification method and verifier, called **M**ulti-**N**euron Constraint Guided **BaB** (MN-BAB), which aims to combine the best of both worlds: it leverages the tight multi-neuron constraints proposed by Müller et al. (2022) within a BaB framework to yield an efficient GPU-based dual method.

The key insight is that the significantly increased precision of the underlying bounding method substantially reduces the number of domain splits (carrying exponential cost) required to certify a property. This improvement is especially pronounced for larger and less regularized networks where additional splits of the problem domain yield diminishing returns. We release all code and scripts to reproduce our experiments at `https://github.com/eth-sri/mn-bab`.

**Main Contributions**:

- We present a novel verification framework, MN-BAB, which leverages tight multi-neuron constraints and a GPU-based dual solver in a BaB approach.
- We develop a novel branching heuristic, ACS, based on information obtained from analyzing our multi-neuron constraints.
- We propose a new class of branching heuristics, CAB, applicable to all BaB-based verifiers, that correct the expected bound improvement of a branching decision for the incurred computational cost.
- Our extensive empirical evaluation demonstrates that we improve on the state of the art in terms of certified accuracy by as much as 28% on challenging networks.

## 2 BACKGROUND

In this section, we review the necessary background for our method (discussed next).

### 2.1 NEURAL NETWORK VERIFICATION

The neural network verification problem can be defined as follows: given a network $f : \mathcal{X} \to \mathcal{Y}$, an input region $\mathcal{D} \subseteq \mathcal{X}$, and a linear property $\mathcal{P} \subseteq \mathcal{Y}$ over the output neurons $y \in \mathcal{Y}$, prove $f(\boldsymbol{x}) \in \mathcal{P}$, $\forall \boldsymbol{x} \in \mathcal{D}$. We instantiate this problem with the challenging $\ell_\infty$-norm bounded perturbations and set $\mathcal{D}$ to the $\ell_\infty$ ball around an input point $\boldsymbol{x}_0$ of radius $\epsilon$: $\mathcal{D}_\epsilon(\boldsymbol{x}_0) = \{\boldsymbol{x} \in \mathcal{X} \mid ||\boldsymbol{x} - \boldsymbol{x}_0||_\infty \leq \epsilon\}$.

For ease of presentation, we consider neural networks of $L$ fully-connected layers with ReLU activation functions (we note that MN-BAB can handle a wide range of layers including convolutional, residual, batch-normalization, and average-pooling layers). We focus on ReLU networks as the BaB framework only yields complete verifiers for piecewise linear activation functions but remark that our approach is applicable to a wide class of activations including ReLU, Sigmoid, Tanh, MaxPool, and others. We denote the number of neurons in the $i^{\text{th}}$ layer as $d_i$ and the corresponding weights and biases as $\boldsymbol{W}^{(i)} \in \mathbb{R}^{d_i \times d_{i-1}}$ and $\boldsymbol{b}^{(i)} \in \mathbb{R}^{d_i}$ for $i \in \{1, ..., L\}$. Further, the neural network is defined as $f(\boldsymbol{x}) := \hat{\boldsymbol{z}}^{(L)}(\boldsymbol{x})$ where $\hat{\boldsymbol{z}}^{(i)}(\boldsymbol{x}) := \boldsymbol{W}^{(i)}\boldsymbol{z}^{(i-1)}(\boldsymbol{x}) + \boldsymbol{b}^{(i)}$ and $\boldsymbol{z}^{(i)}(\boldsymbol{x}) := \max(0, \hat{\boldsymbol{z}}^{(i)}(\boldsymbol{x}))$. For readability, we omit the dependency of intermediate activations on $\boldsymbol{x}$.

Since we can encode any linear property over output neurons into an additional affine layer, we can simplify the general formulation $f(\boldsymbol{x}) \in \mathcal{P}$ to $f(\boldsymbol{x}) > \boldsymbol{0}$. The property can now be verified by proving that a lower bound to the following optimization problem is greater 0:

$$\min_{\boldsymbol{x} \in \mathcal{D}_\epsilon(\boldsymbol{x}_0)} \quad f(\boldsymbol{x}) = \hat{\boldsymbol{z}}^{(L)}$$
$$s.t. \quad \hat{\boldsymbol{z}}^{(i)} = \boldsymbol{W}^{(i)}\boldsymbol{z}^{(i-1)} + \boldsymbol{b}^{(i)} \tag{1}$$
$$\boldsymbol{z}^{(i)} = \max(\boldsymbol{0}, \hat{\boldsymbol{z}}^{(i)})$$

A method is called sound if every property it proves actually holds (no false positives). A method is called complete if it can prove every property that actually holds (no false negatives).

### 2.2 BRANCH-AND-BOUND FOR VERIFICATION

Bunel et al. (2020) successfully applied the Branch-and-Bound (BaB) approach (lan, 1960) to neural network verification. It consists of a bounding method that computes sound upper and lower bounds on the optimization objective of Eq. (1) and a branching method that recursively splits the problem into subproblems with added constraints, allowing for increasingly tighter bounds. If an upper bound (primal solution) $< 0$ is found, this represents a counterexample and allows to terminate the procedure. If a lower bound $> 0$ is obtained, the property is verified on the corresponding (sub-)domain.

If a lower bound $> 0$ is derived on all subdomains, the property is verified. An ideal splitting procedure minimizes the total time required for bounding, which is often well approximated by the number of considered subproblems. A simple approach is to split the input domain, however, this is inefficient for high-dimensional input spaces. Splitting a ReLU activation node into its positive and negative phases has been shown to be far more efficient (Bunel et al., 2020) and ultimately yields a complete verifier (Wang et al., 2021). Hence, we focus solely on ReLU branching strategies.

## 2.3 LINEAR CONSTRAINTS

The key challenge in neural network verification Eq. (1) is handling the non-linear activations. Stable ReLUs, i.e., those which we can show to be always active ($\hat{z} \geq 0$) or inactive ($\hat{z} \leq 0$), can be replaced by linear functions. Unstable ReLUs, i.e., those that can be either active or inactive depending on the exact $x \in \mathcal{D}$, have to be approximated using a convex relaxation of their input-output set. We build on the convex relaxation introduced in DEEPPOLY (Singh et al., 2019b) and shown in Fig. 1. Its key property is the single linear upper and lower bound, which allows for efficient bound computation.

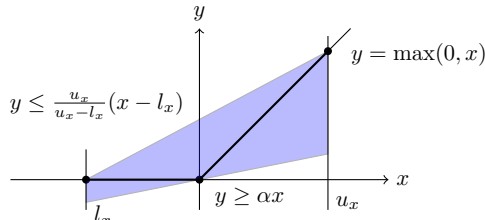

Figure 1: Illustration of the DEEP-POLY relaxation of a ReLU activation $y = \max(x, 0)$ given the neuron-wise bounds $x \in [l_x, u_x]$ and parametrized by $\alpha \in [0, 1]$.

## 2.4 MULTI-NEURON CONSTRAINTS

All convex relaxations that consider ReLU neurons individually are fundamentally limited in their precision by the so-called (single-neuron) convex relaxation barrier (Salman et al., 2019). It can be overcome by considering multiple neurons jointly (Singh et al., 2019a; Tjandraatmadja et al., 2020; Müller et al., 2022; Palma et al., 2021), thereby capturing interactions between these neurons and obtaining tighter bounds, illustrated in Fig. 2. We leverage the multi-neuron constraints from Müller et al. (2022), expressed as a conjunction of linear constraints over the joint input and output space of a ReLU layer.

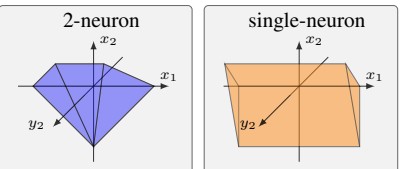

Figure 2: Comparison of multi-neuron and single-neuron constraints projected into $y_2$-$x_1$-$x_2$-space. Reproduced from Müller et al. (2022).

## 2.5 CONSTRAINED OPTIMIZATION VIA LAGRANGE MULTIPLIERS

To express constraints as part of the optimization problem, we use the technique of Lagrange multipliers. Given a constrained minimization problem $\min_{\boldsymbol{x}} f(\boldsymbol{x})$, $s.t.$ $\boldsymbol{g}(\boldsymbol{x}) \leq \boldsymbol{0}$, we can bound the objective function with:

$$\min_{\boldsymbol{x}} f(\boldsymbol{x}) \geq \min_{\boldsymbol{x}} \max_{\boldsymbol{\lambda} \geq \boldsymbol{0}} f(\boldsymbol{x}) + \boldsymbol{\lambda} \boldsymbol{g}(\boldsymbol{x})$$

If a constraint is satisfied, i.e., $g(\boldsymbol{x})_j \leq 0$, $\lambda_j = 0$ maximizes the (inner) objective, else, i.e., $g(\boldsymbol{x})_j > 0$, increasing $\lambda_j$ allows the objective to grow unboundedly, shifting the minimum over $\boldsymbol{x}$ until the constraint is satisfied. Hence, if $\lambda_j > 0$ after optimization, the constraint is active, i.e., it is actively enforced and currently satisfied with equality.

## 3 A MULTI-NEURON RELAXATION BASED BAB FRAMEWORK

In this section, we describe the two key components of MN-BAB: (i) an efficient bounding method leveraging multi-neuron constraints, as well as constrained optimization via lagrange multipliers (§3.1), and (ii) a branching method tailored to it (§3.2).

## 3.1 EFFICIENT MULTI-NEURON BOUNDING

We build on the approach of Singh et al. (2019b), extended by Wang et al. (2021) of deriving a lower bound $\underline{f}$ as a function of the network inputs and a set of optimizable parameters. Crucially, we tighten these relatively loose bounds significantly by enforcing precise multi-neuron constraints

via Lagrange multipliers. To enable this, we develop a method capable of integrating any linear constraint over arbitrary neurons anywhere in the network into the optimization objective. At a high level, we derive linear upper and lower bounds of the form $\boldsymbol{z}^{(i)} \lessgtr \boldsymbol{a}\boldsymbol{z}^{(i-1)} + c$ for every layer's output in terms of its inputs $\boldsymbol{z}^{(i-1)}$. Then, starting with a linear expression in the last layer's outputs $\boldsymbol{z}^{(L)}$ which we aim to bound, we use the linear bounds derived above, to replace $\boldsymbol{z}^{(L)}$ with symbolic bounds depending only on the previous layer's values $\boldsymbol{z}^{(L-1)}$. We proceed in this manner recursively until we obtain an expression only in terms of the networks inputs. Below, we describe this backsubstitution process for ReLU and affine layers.

**Affine Layer** Assume any affine layer $\hat{\boldsymbol{z}}^{(i)} = \mathbf{W}^{(i)}\boldsymbol{z}^{(i-1)} + \mathbf{b}^{(i)}$ and a lower bound $\underline{f} = \hat{\boldsymbol{a}}^{(i)}\hat{\boldsymbol{z}}^{(i)} + \hat{c}^{(i)}$ with respect to its outputs. We then substitute the affine expression for $\hat{\boldsymbol{z}}^{(i)}$ to obtain:

$$\underline{f} = \underbrace{\hat{\boldsymbol{a}}^{(i)}\mathbf{W}^{(i)}}_{\boldsymbol{a}^{(i-1)}}\boldsymbol{z}^{(i-1)} + \underbrace{\hat{\boldsymbol{a}}^{(i)}\mathbf{b}^{(i)} + \hat{c}^{(i)}}_{c^{(i-1)}} = \boldsymbol{a}^{(i-1)}\boldsymbol{z}^{(i-1)} + c^{(i-1)} \tag{2}$$

**ReLU Layer** Let $\underline{f} = \boldsymbol{a}^{(i)}\boldsymbol{z}^{(i)} + c^{(i)}$ be a lower bound with respect to the output of a ReLU layer $\boldsymbol{z}^{(i)} = \max(0, \hat{\boldsymbol{z}}^{(i)})$ and $\mathbf{l}^{(i)}$ and $\mathbf{u}^{(i)}$ bounds on its input s.t. $\mathbf{l}^{(i)} \leq \hat{\boldsymbol{z}}^{(i)} \leq \mathbf{u}^{(i)}$, obtained by recursively applying this bounding procedure or using a cheaper but less precise bounding method (Singh et al., 2018; Gowal et al., 2018). The backsubstitution through a ReLU layer now consists of three distinct steps: 1) enforcing multi-neuron constraints, 2) enforcing single-neuron constraints to replace the dependence on $\boldsymbol{z}^{(i)}$ by $\hat{\boldsymbol{z}}^{(i)}$, and 3) enforcing split constraints, which we describe in detail below.

**Enforcing Multi-Neuron Constraints** We compute multi-neuron constraints as described in Müller et al. (2022), although our approach is applicable to any linear constraints in the input-output space of ReLU activations, written as:

$$\begin{bmatrix}\boldsymbol{P}^{(i)} & \hat{\boldsymbol{P}}^{(i)} & -\mathbf{p}^{(i)}\end{bmatrix}\begin{bmatrix}\boldsymbol{z}^{(i)} \\ \hat{\boldsymbol{z}}^{(i)} \\ 1\end{bmatrix} \leq 0. \tag{3}$$

where $\hat{\boldsymbol{z}}^{(i)}$ are the pre- and $\boldsymbol{z}^{(i)}$ the post-activation values and $\boldsymbol{P}^{(i)}$, $\hat{\boldsymbol{P}}^{(i)}$, and $-\mathbf{p}^{(i)}$ the constraint parameters. We enforce these constraints using Lagrange multipliers (see §2.5), yielding sound lower bounds for all $\boldsymbol{\gamma}^{(i)} \in (\mathbb{R}^{\geq 0})^{e_i}$, where $e_i$ denotes the number of multi-neuron constraints in layer $i$.

$$\boldsymbol{a}^{(i)}\boldsymbol{z}^{(i)} + c^{(i)} \geq \max_{\boldsymbol{\gamma}^{(i)} \geq 0} \boldsymbol{a}^{(i)}\boldsymbol{z}^{(i)} + c^{(i)} + \boldsymbol{\gamma}^{(i)\top}(\boldsymbol{P}^{(i)}\boldsymbol{z}^{(i)} + \hat{\boldsymbol{P}}^{(i)}\hat{\boldsymbol{z}}^{(i)} - \mathbf{p}^{(i)})$$

$$= \max_{\boldsymbol{\gamma}^{(i)} \geq 0} \underbrace{(\boldsymbol{a}^{(i)} + \boldsymbol{\gamma}^{(i)\top}\boldsymbol{P}^{(i)})}_{\boldsymbol{a}'^{(i)}}\boldsymbol{z}^{(i)} + \underbrace{\boldsymbol{\gamma}^{(i)\top}\hat{\boldsymbol{P}}^{(i)}\hat{\boldsymbol{z}}^{(i)} + \boldsymbol{\gamma}^{(i)\top}(-\mathbf{p}^{(i)}) + c^{(i)}}_{c'^{(i)}}$$

Note that this approach can be easily extended to linear constraints over any activations in arbitrary layers if applied in the last affine layer at the very beginning of the backsubstitution process.

**Enforcing Single-Neuron Constraints** We now apply the single-neuron DEEPPOLY relaxation with parametrized slopes $\alpha$ collected in $\boldsymbol{D}$ (see below):

$$\max_{\boldsymbol{\gamma}^{(i)} \geq 0} \boldsymbol{a}'^{(i)}\boldsymbol{z}^{(i)} + c'^{(i)} \geq \max_{\substack{0 \leq \boldsymbol{\alpha}^{(i)} \leq 1 \\ \boldsymbol{\gamma}^{(i)} \geq 0}} \boldsymbol{a}'^{(i)}(\boldsymbol{D}^{(i)}\hat{\boldsymbol{z}}^{(i)} + \underline{\boldsymbol{b}}^{(i)}) + c'^{(i)}$$

$$= \max_{\substack{0 \leq \boldsymbol{\alpha}^{(i)} \leq 1 \\ \boldsymbol{\gamma}^{(i)} \geq 0}} \underbrace{\boldsymbol{a}'^{(i)}\boldsymbol{D}^{(i)}}_{\boldsymbol{a}''^{(i)}}\hat{\boldsymbol{z}}^{(i)} + \underbrace{\boldsymbol{a}'^{(i)}\underline{\boldsymbol{b}}^{(i)} + c'^{(i)}}_{c''^{(i)}}$$

The intercept vector $\underline{\boldsymbol{b}}$ and the diagonal slope matrix $\boldsymbol{D}$ are defined as:

$$D_{j,j} = \begin{cases} 1 & \text{if } l_j \geq 0 \text{ or node } j \text{ is positively split} \\ 0 & \text{if } u_j \leq 0 \text{ or node } j \text{ is negatively split} \\ \alpha_j & \text{if } l_j < 0 < u_j \text{ and } a_j \geq 0 \\ \frac{u_j}{u_j - l_j} & \text{if } l_j < 0 < u_j \text{ and } a_j < 0 \end{cases}$$

$$\underline{b}_j = \begin{cases} -\frac{u_j l_j}{u_j - l_j} & \text{if } l_j < 0 < u_j \text{ and } a_j < 0 \\ 0 & \text{otherwise} \end{cases}$$

Where we drop the layer index $i$ for readability and $\alpha_j$ is the lower bound slope parameter illustrated in Fig. 1. Note how, depending on whether the sensitivity $a_j^{(i)}$ of $\underline{f}$ w.r.t. $z_j^{(i)}$ has positive or negative sign, we substitute $z_j^{(i)}$ for its lower or upper bound, respectively.

**Enforcing Split Constraints** We encode split constraints of the form $\hat{z}_j^{(i)} \geq 0$ or $\hat{z}_j^{(i)} \leq 0$ using the diagonal split matrix $S$ as follows, again dropping the layer index $i$:

$$S_{j,j} = \begin{cases} -1 & \text{positive split: } \hat{z}_j \geq 0 \\ 1 & \text{negative split: } \hat{z}_j < 0 \\ 0 & \text{no split} \end{cases}$$

$$S^{(i)} \hat{z}^{(i)} \leq 0$$

We again enforce these constraints using Lagrange multipliers:

$$\max_{\substack{0 \leq \boldsymbol{\alpha}^{(i)} \leq 1 \\ \boldsymbol{\gamma}^{(i)} \geq 0}} \boldsymbol{a}''^{(i)} \hat{z}^{(i)} + c''^{(i)} \geq \max_{\substack{0 \leq \boldsymbol{\alpha}^{(i)} \leq 1 \\ \boldsymbol{\beta}^{(i)} \geq 0 \\ \boldsymbol{\gamma}^{(i)} \geq 0}} \underbrace{(\boldsymbol{a}''^{(i)} + \boldsymbol{\beta}^{(i)\top} S^{(i)})}_{\boldsymbol{a}'''^{(i)}} \hat{z}^{(i)} + \underbrace{c''^{(i)}}_{c'''^{(i)}}$$

Putting everything together, the backsubstitution operation through a ReLU layer is:

$$\min_{x \in \mathcal{D}} \boldsymbol{a}^{(i)} z^{(i)} + c^{(i)} \geq \min_{x \in \mathcal{D}} \max_{\substack{0 \leq \boldsymbol{\alpha}^{(i)} \leq 1 \\ \boldsymbol{\beta}^{(i)} \geq 0 \\ \boldsymbol{\gamma}^{(i)} \geq 0}} \underbrace{((\boldsymbol{a}^{(i)} + \boldsymbol{\gamma}^{(i)\top} P^{(i)}) D^{(i)} + \boldsymbol{\beta}^{(i)\top} S^{(i)} + \boldsymbol{\gamma}^{(i)\top} \hat{P}^{(i)})}_{\hat{\boldsymbol{a}}^{(i)}} \hat{z}^{(i)}$$
$$+ \underbrace{(\boldsymbol{a}^{(i)} + \boldsymbol{\gamma}^{(i)\top} P^{(i)})' \underline{b}^{(i)} + \boldsymbol{\gamma}^{(i)\top} (-\mathbf{p}^{(i)}) + c^{(i)}}_{\hat{c}^{(i)}} \tag{4}$$

Full backsubstitution through all layers leads to an optimizable lower bound on $\underline{f}$:

$$\min_{\boldsymbol{x} \in \mathcal{D}} f(x) \geq \min_{\boldsymbol{x} \in \mathcal{D}} \max_{\substack{0 \leq \boldsymbol{\alpha} \leq 1 \\ 0 \leq \boldsymbol{\beta} \\ 0 \leq \boldsymbol{\gamma}}} \boldsymbol{a}^{(0)} \boldsymbol{x} + c^{(0)} \geq \max_{\substack{0 \leq \boldsymbol{\alpha} \leq 1 \\ 0 \leq \boldsymbol{\beta} \\ 0 \leq \boldsymbol{\gamma}}} \min_{\boldsymbol{x} \in \mathcal{D}} \boldsymbol{a}^{(0)} \boldsymbol{x} + c^{(0)}$$

where the second inequality holds due to weak duality. We denote all $\alpha_j^{(i)}$ from every layer of the backsubstitution process with $\boldsymbol{\alpha}$ and define $\boldsymbol{\beta}$ and $\boldsymbol{\gamma}$ analogously. The inner minimization over $\boldsymbol{x}$ has a closed form solution if $\mathcal{D}$ is an $l_p$-ball of radius $\epsilon$ around $\boldsymbol{x}_0$, given by Hölder's inequality:

$$\max_{\substack{0 \leq \boldsymbol{\alpha} \leq 1 \\ 0 \leq \boldsymbol{\beta} \\ 0 \leq \boldsymbol{\gamma}}} \min_{\boldsymbol{x} \in \mathcal{D}} \boldsymbol{a}^{(0)} \boldsymbol{x} + c^{(0)} \geq \max_{\substack{0 \leq \boldsymbol{\alpha} \leq 1 \\ 0 \leq \boldsymbol{\beta} \\ 0 \leq \boldsymbol{\gamma}}} \boldsymbol{a}^{(0)} \boldsymbol{x}_0 - \|\boldsymbol{a}^{(0)\top}\|_q \epsilon + c^{(0)} \tag{5}$$

where $q$ is defined $s.t.$ $\frac{1}{p} + \frac{1}{q} = 1$. Since these bounds are sound for any $0 \leq \boldsymbol{\alpha} \leq 1$, and $\boldsymbol{\beta}, \boldsymbol{\gamma} \geq 0$, we tighten them by using (projected) gradient ascent to optimize these parameters.

We compute all intermediate bounds using the same bounding procedure, leading to two full parameter sets for every neuron in the network. To reduce memory requirements, we share parameter sets between all neurons in the same layer, but keep separate sets for upper and lower bounds.

**Upper Bounding the Minimum Objective** Showing an upper bound on the minimum optimization objective precluding verification ($\underline{f} < 0$) allows us to terminate the BaB process early. Propagating any input $\boldsymbol{x} \in \mathcal{D}$ through the network yields a valid upper bound, hence, we use the input that minimizes Eq. (5):

$$x_i = \begin{cases} (x_0)_i + \epsilon & \text{if } a_i^{(0)} < 0 \\ (x_0)_i - \epsilon & \text{if } a_i^{(0)} \geq 0 \end{cases}.$$

### 3.2 Multi-Neuron Constraint Guided Branching

Generally, the BaB approach is based on recursively splitting an optimization problem into easier subproblems to derive increasingly tighter bounds. However, the benefit of different splits can vary widely, making an effective branching heuristic which chooses splits that minimize the total number of required subproblems a key component of any BaB framework (Bunel et al., 2020). Typically, a score is assigned based on the expected bound improvement and the most promising split is chosen. Consequently, the better this score captures the actual bound improvement, the better the resulting decision. Both the commonly used BABSR (Bunel et al., 2020) and the more recent FSB (De Palma et al., 2021) approximate improvements under a DEEPPOLY style backsubstitution procedure. As neither considers the impact of multi-neuron constraints, the scores they compute might not be suitable proxies for the bound improvements obtained with our method. To overcome this issue, we propose a novel branching heuristic, *Active Constraint Score Branching* (ACS), considering multi-neuron constraints. Further, we introduce a branching heuristic framework which corrects the expected bound improvement with the potentially significantly varying expected computational cost.

**Active Constraint Score Branching**   The value of a Lagrange parameter $\gamma$ provides valuable information about the constraint it enforces. Concretely, $\gamma > 0$ indicates that a constraint is active, i.e., the optimal solution fulfills it with equality. Further, for a constraint $g(x) \leq 0$, a larger $\partial_x \gamma g(x)$ indicates a larger sensitivity of the final bound to violations of this constraint. We compute this sensitivity for our multi-neuron constraints with respect to both ReLU outputs and inputs as $\gamma^\top P$ and $\gamma^\top \hat{P}$, respectively, where $P$ and $\hat{P}$ are the multi-neuron constraint parameters. We then define our branching score for a neuron $j$ in layer $i$ as the sum over all sensitivities with respect to its input or output:

$$s_{i,j} = |\gamma^{(i)\top} P^{(i)}|_j + |\gamma^{(i)\top} \hat{P}^{(i)}|_j, \tag{6}$$

Intuitively, splitting the node with the highest cumulative sensitivity makes its encoding exact and effectively tightens all of these high sensitivity constraints. We can efficiently compute those scores without an additional backsubstitution pass.

**Cost Adjusted Branching**   Any complete method will decide every property eventually. Hence, its runtime is a key performance metric. Existing branching heuristics ignore this aspect and only consider the expected improvement in bound-tightness, but not the sometimes considerable differences in computational cost. We propose *Cost Adjusted Branching* (CAB), scaling the expected bound improvement (approximated with the branching score) with the inverse of the cost expected for the split, and then picking the branching decision yielding the highest expected bound improvement per cost. The true cost of a split consists of the direct cost of the next branching step and the change of cumulative cost for all consecutive steps. We find a local approximation considering just the former component, similar to only considering the one-step bound improvement, to be a good proxy. We approximate this direct cost by the number of floating-point operations required to compute the new bounds, refer to App. B for a more detailed description. Note that any approximation of the expected cost can be used to instantiate CAB.

**Enforcing Splits**   Once the ReLU node to split on has been determined, two subproblems are generated. One where a non-negative $\hat{z} \geq 0$ and one where non-positive $\hat{z} \leq 0$ pre-activation value has to be enforced. This is accomplished by setting the corresponding entries in the split matrix $S$, used during the bounding process, to $-1$ and $1$, respectively. As the intermediate bounds for all layers up to and including the one that was split remain unchanged, we do not recompute them.

### 3.3 Soundness and Completeness

The soundness of the BaB approach follows directly from the soundness of the underlying bounding method discussed in Section 3.1. To show completeness, it is sufficient to consider the limit case where all ReLU nodes are split and the network becomes linear, making DEEPPOLY relaxations exact. To also obtain exact bounds, all split constraints have to be enforced. This can be done by computing optimal $\beta$ for the now convex optimization problem (Wang et al., 2021). It follows that a property holds if and only if the exact bounds thus obtained are positive on all subproblems. We conclude that MN-BAB is a complete verifier.

Table 1: Natural accuracy [%] (Acc.), verified accuracy [%] (Ver.) and its empirical upper bound [%] and average runtime [s] of the first 1000 images of the test set.

| Dataset | Model | Acc. | $\epsilon$ | ERAN | | OVAL | | $\beta$-CROWN | | MN-BAB (ours) BABSR +CAB | | MN-BAB (ours) ACS +CAB | | Upper Bound |
|---|---|---|---|---|---|---|---|---|---|---|---|---|---|---|
| | | | | Ver. | Time | Ver. | Time | Ver. | Time | Ver. | Time | Ver. | Time | |
| MNIST | ConvSmall | 98.0 | 0.12 | **73.2** | 38.4 | 69.8 | 26.2 | 71.6[†] | 46 | 71.0 | 21.3 | 70.3 | 26.2 | 73.2 |
| | ConvBig | 92.9 | 0.30 | **78.6** | **6.0** | – | – | 77.7[†] | 78 | 78.3 | 20.8 | 77.2 | 46.2 | 78.6 |
| | ConvSuper | 97.7 | 0.18 | 0.5 | 142.0 | – | – | – | – | **19.2** | **86.2** | 17.6 | 90.9 | 37.3 |
| CIFAR10 | ConvSmall | 63.0 | 2/255 | **47.2** | 54.4 | 46.2 | 17.7 | 46.3[†] | 18 | 46.1 | 16.4 | 45.8 | 18.0 | 48.1 |
| | ConvBig | 63.1 | 2/255 | 48.2 | 128.1 | 50.6 | 42.0 | 50.3[†] | 55 | 49.4 | 49.5 | **51.5** | **37.0** | 55.0 |
| | ResNet6-A [‡] | 84.0 | 1/255 | 45.0 | 114.6 | • | • | 52.0 | 263.4 | 48.0 | 202.7 | **55.0** | **170.7** | 75.0 |
| | ResNet6-B [‡] | 79.0 | 1/255 | 66.0 | 48.6 | • | • | **67.0** | 105.7 | 65.0 | 51.1 | **67.0** | **37.8** | 71.0 |
| | ResNet8-A [‡] | 83.0 | 1/255 | 11.0 | 362.5 | • | • | 18.0 | 497.3 | 19.0 | 390.7 | **23.0** | 371.0 | 70.0 |

[†] We report numbers from Wang et al. (2021). − Errors prevent reporting reliable numbers. [‡] Due to long runtimes, we evaluated only on the first 100 samples of the test set. • OVAL does not support ResNet architectures.

# 4 EXPERIMENTAL EVALUATION

We now present an extensive evaluation of our method. First, and perhaps surprisingly, we find that existing MILP-based verification tools (Singh et al., 2019c; Müller et al., 2022) are more effective in verifying robustness on many established benchmark networks than what is considered state-of-the-art. This highlights that next-generation verifiers should focus on and be benchmarked using less regularized and more accurate networks. We take a step in this direction by proposing and comparing on such networks, before analyzing the effectiveness of the different components of MN-BAB in an extensive ablation study.

**Experimental Setup** We implement a GPU-based version of MN-BAB in PyTorch (Paszke et al., 2019) and evaluate all benchmarks using a single NVIDIA RTX 2080Ti, 64 GB of memory, and 16 CPU cores. We attempt to falsify every property with an adversarial attack, before beginning certification. For every subproblem, we first lower-bound the objective using DEEPPOLY and then compute refined bounds using the method described in §3.

**Benchmarks** We benchmark MN-BAB on a wide range of networks (see Table 3 in App. A) on the MNIST (Deng, 2012) and CIFAR10 datasets (Krizhevsky et al., 2009). We also consider three new residual networks, ResNet6-A, ResNet6-B, and ResNet8-A. ResNet6-A and ResNet6-B have the same architecture but differ in regularization strength while ResNet8-A has an additional residual block. All three were trained with adversarial training (Madry et al., 2018) using PGD, the GAMA loss (Sriramanan et al., 2020) and MixUp data augmentation (Zhang et al., 2021). ResNet6-A and ResNet8-A were trained using 8-steps and $\epsilon = 4/255$, whereas 20-steps and $\epsilon = 8/255$ were used for ResNet6-B. We compare against $\beta$-CROWN (Wang et al., 2021), a BaB-based state-of-the-art complete verifier, OVAL (Palma et al., 2021; De Palma et al., 2021; Bunel et al., 2020), a BaB framework based on a different class of multi-neuron constraints, and ERAN Singh et al. (2019c); Müller et al. (2022) combining the incomplete verifier PRIMA, whose tight multi-neuron constraints we leverage in our method, and a (partial) MILP encoding.

**Comparison to State-of-the-Art Methods** In Table 1, we compare the verified accuracy and runtime of MN-BAB with that of state-of-the-art tools ERAN, $\beta$-CROWN, and OVAL. Perhaps surprisingly, we find that both MNIST and the smallest CIFAR10 benchmark network established over the last years (Singh et al., 2019b) can be verified completely or almost completely in less than 50s per sample using ERAN, making them less relevant as benchmarks for new verification tools. On these networks, all three BaB methods (including ours) are outperformed to a similar degree, with the reservation that we could not evaluate OVAL on MNIST ConvBig due to runtime errors. On the remaining unsolved networks where complete verification via a MILP encoding does not scale, MN-BAB consistently obtains the highest certified accuracy and for all but one also the lowest runtime. On ConvSuper we were not able to find configurations for OVAL and $\beta$-CROWN that did not run out of memory. Comparing the BaB methods, we observe an overall trend that more precise but also expensive underlying bounding methods are most effective on larger networks, where additional splits are less efficient, and complex inter-neuron interactions can be captured by the precise multi-neuron constraints. On these networks, the more complex interactions also lead to more active Multi-Neuron Constraints (MNCs) and hence more informative ACS scores.

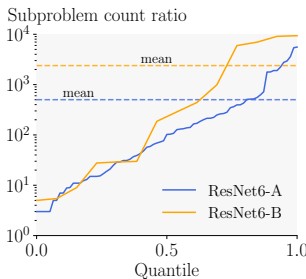

Figure 3: Ratio of subproblems required per property without and with MNCs.

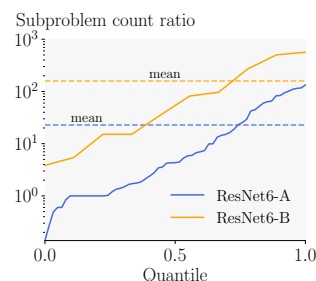

Figure 4: Ratio of subproblems required per property with BABSR and ACS.

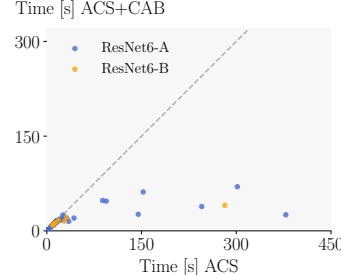

Figure 5: Effect of Cost Adjusted Branching on mean verification time with ACS.

**Ablation Study** To analyze the individual components of MN-BaB, we consider the weakly and heavily regularized `ResNet6-A` and `ResNet6-B`, respectively, with identical architecture. Concretely, we show the effect different bounding procedures and branching approaches have in the two settings in Table 2. As verified accuracy and mean runtime are very coarse performance metrics, we also analyze the ratio of runtimes and number of subproblems required for verification on a per-property level, filtering out those where both methods verify before any branching occurred (Figs. 3–5). Overall, we observe that the number of visited subproblems required for certification can be reduced by two to three orders of magnitude by leveraging precise multi-

Table 2: Verified accuracy [%] (Ver.) and avg. runtime [s] on the first 100 images of the test set for $\epsilon = 1/255$.

| Model | Acc. | Upper Bound | Branching Method | MNCs | MN-BAB | |
| --- | --- | --- | --- | --- | --- | --- |
| | | | | | Ver | Time |
| `ResNet6-A` | 84 | 75 | No Branching | no | 30 | 0.4 |
| | | | No Branching | yes | 39 | 13.2 |
| | | | BABSR | no | 42 | 247.1 |
| | | | BABSR +CAB | no | 46 | 219.3 |
| | | | FSB | yes | 45 | 239.7 |
| | | | BABSR | yes | 47 | 212.8 |
| | | | BABSR +CAB | yes | 48 | 202.7 |
| | | | ACS | yes | 51 | 186.4 |
| | | | ACS +CAB | yes | **55** | 170.7 |
| `ResNet6-B` | 79 | 71 | No Branching | no | 58 | 0.7 |
| | | | No Branching | yes | 61 | 4.1 |
| | | | BABSR | no | 61 | 78.3 |
| | | | BABSR +CAB | no | 63 | 64.9 |
| | | | FSB | yes | 64 | 67.0 |
| | | | BABSR | yes | 63 | 65.5 |
| | | | BABSR +CAB | yes | 65 | 51.1 |
| | | | ACS | yes | 65 | 52.0 |
| | | | ACS +CAB | yes | **67** | 37.8 |

neuron constraints and then again by another one to two orders of magnitude by our novel branching heuristic, ACS. Our cost-adjusted branching yields an additional speed up of around 50%.

The trend of MN-BAB succeeding on more challenging networks is confirmed here. Leveraging MNCs enables us to verify 20% more samples while being around 22% faster (see Table 2) on `ResNet6-A` while on the more heavily regularized `ResNet6-B` we only verify 6% more samples. When analyzing on a per-property level, shown in Fig. 6, the trend is even more pronounced. For easy problems, leveraging MNCs and ACS has little impact (points in the lower left-hand corner). However, it completely dominates on the harder properties where only using single-neuron constraints and BABSR takes up to 33 times longer (points below the diagonal).

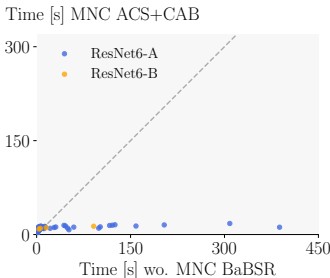

Figure 6: Per property verification times using MN-BAB over those without MNCs and using BABSR.

**Effectiveness of Multi-Neuron Constraints** In Fig. 3, we show the ratio between the number of subproblems required to prove the same lower bound on the final objective (either 0, if both methods certify, or the smaller of the two lower bounds at termination) with and without MNCs over the quantile of properties for `ResNet6-A` (blue) and `ResNet6-B` (orange). We observe that using MNCs reduces the number of subproblems for both networks by between two and three orders of magnitude. Despite the higher per bounding-step cost, this leads to the use of MNCs increasing the number of verified samples by up to 12% while reducing average certification times (see Table 2).

**Effectiveness of ACS Branching** In Fig. 4, we show the ratio between the number of subproblems considered when using BABSR vs. ACS over the quantile of properties. We observe that ACS yields significantly fewer subproblems on most (75%) or all properties on `ResNet6-A` and

`ResNet6-B`, respectively, leading to an additional reduction by between one and two orders of magnitude and showing the effectiveness of our novel ACS branching heuristic. Average verification times are reduced by $12\%$ and $21\%$ on `ResNet6-A` and `ResNet6-B`, respectively. Note that the relatively small improvements in timings are due to timeouts for both methods yielding equally high runtimes which dominate the mean. FSB is consistently outperformed by ACS, certifying $12\%$ less samples on `ResNet6-A`.

**Effectiveness of Cost Adjusted Branching**   In Fig. 5, we show the per property verification times with ACS + CAB over those with ACS. Using CAB is faster (points below the dashed line) for all properties, sometimes significantly so, leading to an average speedup of around $50\%$. Analyzing the results in Table 2, we observe that CAB is particularly effective in combination with the ACS scores and multi-neuron constraints, where bounding costs vary more significantly.

## 5   RELATED WORK

**Neural Network Verification Beyond the Single Neuron Convex Barrier**   After the so-called (Single Neuron) Convex Barrier has been described by Salman et al. (2019) for incomplete relaxation-based methods, a range of approaches has been proposed that consider multiple neurons jointly to obtain tighter relaxations. Singh et al. (2019a) and Müller et al. (2022) derive joint constraints over the input-output space of groups of neurons and refine their relaxation using the intersection of these constraints. Tjandraatmadja et al. (2020) merge the ReLU and preceding affine layer to consider multiple inputs but only one output at a time. While the two approaches are theoretically incomparable, the former yields empirically better results (Müller et al., 2022).

Early complete verification methods relied on off-the-shelf SMT (Katz et al., 2017; Ehlers, 2017) or MILP solvers (Dutta et al., 2018; Tjeng et al., 2019). However, these methods do not scale beyond small networks. In order to overcome these limitations, Bunel et al. (2020) formulated a BaB style framework for complete verification and showed it contains many previous methods as special cases. The basic idea is to recursively split the verification problem into easier subproblems on which cheap incomplete methods can show robustness. Since then, a range of partially (Xu et al., 2021) or fully (Wang et al., 2021; Palma et al., 2021) GPU-based BaB frameworks have been proposed. The most closely related, Palma et al. (2021), leverages the multi-neuron constraints from Tjandraatmadja et al. (2020) but yields an optimization problem of different structure, as constraints only ever include single output neurons.

**Branching**   Most ReLU branching methods proposed to date use the bound improvement of the two child subproblems as the metric to decide which node to branch on next. Full strong branching (Applegate et al., 1995) exhaustively evaluates this for all possible branching decisions. However, this is intractable for all but the smallest networks. Lu & Kumar (2020) train a GNN to imitate the behavior of full strong branching at a fraction of the cost, but transferability remains an open question and collecting training data to imitate is costly. Bunel et al. (2020) proposed an efficiently computable heuristic score, locally approximating the bound improvement of a branching decision using the method of Wong & Kolter (2018). Henriksen & Lomuscio (2021) refine this approach by additionally approximating the indirect effect of the branching decision, however, this requires using two different bounding procedures. De Palma et al. (2021) introduced filtered-smart-branching (FSB), using BaBSR to select branching candidates and then computing a more accurate heuristic score only for the selected candidates. Instead of targeting the largest bound improvement, Kouvaros & Lomuscio (2021) aim to minimize the number of unstable neurons by splitting the ReLU node with the most other ReLU nodes depending on it.

## 6   CONCLUSION

We propose the complete neural network verifier MN-BAB. Building on the Branch-and-Bound methodology, MN-BAB leverages tight multi-neuron constraints, a novel branching heuristic and an efficient dual solver, able to utilize massively parallel hardware accelerators, to enable the verification of particularly challenging networks. Our thorough empirical evaluation shows how MN-BAB is particularly effective in verifying challenging networks with high natural accuracy and practical relevance, reaching a new state-of-the-art in several settings.

## 7 ETHICS STATEMENT

Most machine learning based systems can be both employed with ethical as well as malicious purposes. Methods such as ours that enable the certification of robustness properties of neural networks are a step towards more safe and trustworthy AI systems and can hence amplify any such usage. Further, malicious actors might aim to convince regulators that the proposed approach is sufficient to show robustness to perturbations encountered during real world application, which could lead to insufficient regulation in safety critical domains.

## 8 REPRODUCIBILITY STATEMENT

We will make all code and trained networks required to reproduce our experiments available during the review process as supplementary material and provide instructions on how to run them. Upon publication, we will also release them publicly. We explain the basic experimental setup in Section 4 and provide more details in Section A. All datasets used in the experiments are publicly available. Random seeds are fixed where possible and provided in the supplementary material.

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

Table 3: Overview of the experimental configuration for every network.

| Dataset | Model | Training | Timeout | Batch sizes | #Activation Layers | #Activation Nodes |
|---------|-------|----------|---------|-------------|--------------------|-------------------|
| MNIST | ConvSmall | NOR | 360 | [100, 200, 400] | 3 | 3 604 |
| | ConvBig | DiffAI | 2000 | [2, 2, 4, 8, 12, 20] | 6 | 34 688 |
| | ConvSuper | DiffAI | 360 | [1, 2, 2, 3, 4, 8] | 6 | 88 500 |
| CIFAR10 | ConvSmall | PGD | 360 | [100, 150, 250] | 3 | 4 852 |
| | ConvBig | PGD | 500 | [3, 3, 6, 6, 8, 16] | 6 | 62 464 |
| | ResNet6-A | PGD | 600 | [4, 4, 8, 12, 16, 100] | 6 | 10 340 |
| | ResNet6-B | PGD | 600 | [4, 8, 32, 64, 128, 256] | 6 | 10 340 |
| | ResNet8-A | PGD | 600 | [2, 2, 2, 4, 8, 16, 32, 64] | 8 | 11 364 |

## A  EXPERIMENT DETAILS

In Table 3 we show the per-sample timeout and the batch sizes that were used in the experiments. The timeouts for the first 4 networks were chosen to approximately match the average runtimes reported by Wang et al. (2021), to facilitate comparability.

Since we keep intermediate bounds of neurons before the split layer fixed, as described in Section 3.2, the memory requirements for splitting at different layers can vary. We exploit this fact and choose batch sizes for our bounding procedure depending on the layer where the split occurred.

In order to falsify properties more quickly, we run a strong adversarial attack with the following parameters before attempting certification: We apply two targeted versions (towards all classes) of PGD (Madry et al., 2018) using margin loss (Gowal et al., 2018) and GAMA loss (Sriramanan et al., 2020), both with 5 restarts, 50 steps, and 10 step output diversification (**?**).

### A.1  ARCHITECTURES

In this section, we provide an overview of all the architectures evaluated in Section 4. The architectures of the convolutional networks for MNIST and CIFAR10 are detailed in Table 4. The architectures of both ResNet6-A and ResNet6-B are given in Table 5

Table 4: Network architectures of the convolutional networks for CIFAR10 and MNIST. All layers listed below are followed by an activation layer. The output layer is omitted. 'CONV c h×w/s/p' corresponds to a 2D convolution with c output channels, an h×w kernel size, a stride of s in both dimensions, and an all-around zero padding of p.

| ConvSmall | ConvBig | ConvSuper |
|-----------|---------|-----------|
| CONV 16 4×4/2/0 | CONV 32 3×3/1/1 | CONV 32 3×3/1/0 |
| CONV 32 4×4/2/0 | CONV 32 4×4/2/1 | CONV 32 4×4/1/0 |
| FC 100 | CONV 64 3×3/1/1 | CONV 64 3×3/1/0 |
| | CONV 64 4×4/2/1 | CONV 64 4×4/1/0 |
| | FC 512 | FC 512 |
| | FC 512 | FC 512 |

## B  SPLIT-COST COMPUTATION FOR CAB

Recall that for CAB, we normalize the branching score obtained with an arbitrary branching heuristic with the (approximate) cost of the corresponding split. The true cost of a split consists of the direct cost of the next branching step and the change of cumulative cost for all consecutive steps. As a local approximation, we just consider the former component.

We approximate this direct cost by the number of floating-point operations required to compute the new bounds. This is computed as the sum of floating-point operations required for bounding all intermediate bounds that are recomputed. Our bounding approach only enforces constraints

Table 5: Network architecture of the `ResNet6` and `ResNet8`. All layers listed below are followed by a ReLU activation layer, except if they are followed by a RESADD layer. The output layer is omitted. 'CONV c h×w/s/p' corresponds to a 2D convolution with c output channels, an h×w kernel size, a stride of s in both dimensions, and an all-around zero padding of p.

| ResNet6 | | ResNet8 | |
|---|---|---|---|
| CONV 16 3×3/1/1 | | CONV 16 3×3/2/1 | |
| CONV 32 1×1/2/0 | CONV 32 3×3/2/1 | CONV 32 1×1/2/0 | CONV 32 3×3/2/1 |
| | CONV 32 3×3/1/1 | | CONV 32 3×3/1/1 |
| RESADD | | RESADD | |
| CONV 64 1×1/2/0 | CONV 64 3×3/2/1 | CONV 64 1×1/2/0 | CONV 64 3×3/2/1 |
| | CONV 64 3×3/1/1 | | CONV 64 3×3/1/1 |
| RESADD | | RESADD | |
| FC 100 | | CONV 128 1×1/2/0 | CONV 128 3×3/2/1 |
| | | | CONV 128 3×3/1/1 |
| | | RESADD | |
| | | FC 100 | |

on neurons preceding the neurons included in the bounding objective. Hence, we only recompute intermediate bounds for layers after the layer where the split occurs, as discussed in §3.2.

To compute the cost of recomputing the lower (or upper) bound of one intermediate node, we add up all floating point operations needed to perform the backsubstitution described in §3.1. As backsubstitution is just a series of matrix multiplications, the number of required floating point operations can be deduced from the sizes of the multiplied matrices.

Thus if we split on layer $k$ of an $L$ layer network and the cost of backsubstituion from layer $i$ is $C_i$ and the number of nodes is $d_i$, the final cost of the split is:

$$\sum_{i=k+1}^{L} 2d_i C_i$$

Where the factor 2 comes from the fact that for intermediate bounds we need to recompute both lower and upper bounds. The cost $C_i$ of a full backsubstitution from layer $i$ can be computed as the sum over the cost of backsubstituting through all preceding layers $C_i = \sum_{j=0}^{i-1} c_j$, where the cost for a single layer can be computed as follows:

- ReLU layer: $c_j = d_j + p_j$, where $p_j$ is the number of multi-neuron constraints.
- Linear layer: $c_j = \#W_j$, where $\#W_j$ is the number of elements in the weight matrix.
- Conv layer: $c_j = d_j k_j^2$, where $k_j$ is the kernel size.

