# OpenReview forum: "Complete Verification via Multi-Neuron Relaxation Guided Branch-and-Bound"
_ICLR.cc/2022/Conference — ICLR 2022 Poster_

### Official Review · Reviewer_3m7j · 2021-11-01

**Correctness:** 4
**Technical Novelty And Significance:** 2
**Empirical Novelty And Significance:** Not applicable
**Recommendation:** 6
**Confidence:** 5

**Main Review:**

# Strengths of the paper:
The paper is easy to follow, and provide clear attributions of the ideas it is based on, citing the relevant works. The approach suggested makes sense and the authors have indicated that they will release the code to reproduce their results.
The empirical results that are included are useful to assess how much methods like beta crown can be extended to tighter relaxations.

# Clarification needed
## Q1: Clarification about Cost Adjusted Branching
The description of CAB is that it "scales the expected bound improvement with the inverse of the cost expected for the split".
There is no description about how "the expected bound improvement" is computed. If this the score given by a method like "Active Constraint Score Branching" or BABSR, I think it's inaccurate to call it like this.
There should be a clarification as to why different branching decisions have "considerable differences in computational cost". Is it simply because branching a "late" layer requires to recompute less intermediate bounds? This is unclear because nowhere is described what exactly you compute at each level (you could decide to tighten intermediate bounds based on the new branching constraints, even if they appear before the branching constraint)

## Q2: Experimental evaluation not supporting the claim
The last contribution listed is that the authors "improve on the state-of-the-art in terms of certified accuracy by as much as 26% while being 30% faster". This numbers come from the experiments that are listed in Table 2, but this is only a comparison to different ablated version of the authors method. I understand that beta-crown would be equivalent to (BaBSR, no MNC) but this is still missing a comparison to other methods that use tighter relaxation than single neuron relaxation. Even if beta-crown and OVAL do not support residual connections, why not train a larger CNN if the issues of the models of table 1 is that the network are too easy?
Similarly, the only baseline used for Branching method proposed is BaBSR, while something like Filtered Smart Branching (FSB) could have been tested (the authors mention that method in their related works)

When compared against strong baselines (Table 1), the improvements of the method proposed are significantly lower.

In addition, is there a reason why for Table 1, the branching is BaBSR+CAB for all but one of the networks, rather than ACS + CAB, given than ACS is one of the paper's contribution?

There is also the additional problems of comparing different branch and bound verifiers on only two numbers. With branch and bound verifiers, It is always possible to obtain higher verified accuracy if you are willing to trade off time (just do more branching steps). Comparing two methods based on only the verified accuracy and the time it took means that there is only a partial view and the result might depend on some settings with regards to termination. Why not follow the usual method of drawing cactus plots as is common in complete verification paper?

# Small questions:
- SQ1: Why is the verification attempt on MNIST in Table 1 given for eps=0.120? I am familiar with seeing verification for eps=0.1 or eps=0.3. Is there a motivation for departing from standard benchmarks?


**Summary Of The Paper:**

This paper deals with complete verification of neural network, based on a branch and bound framework.

It is essentially a combination of two recent papers:
- Beta-crown[1], which optimizes with first order methods the bounds obtained by LIRPA type bounds. Additional constraints on the network activations (such as the ones resulting from branching) are included through lagrange multipliers and optimized over.
- Prima[2], which derives linear constraints to impose on the activation of the network to obtain tighter relaxations. These constraints comes from the fact that you can obtain a tighter convex hull by relaxing a group of neurons together, rather than relaxing each neuron independently. In the Prima paper, the constraints are simply encoded into an off-the-shelf LP solver.

The main contribution of this paper comes from adding the linear constraints that comes from the Prima relaxation (using Lagrange multiplier) to the problem optimized by Beta-crown.

In addition to that, the paper also propose new branching heuristics:
- Active Constraint Score Branching, which uses the lagrangian variables associated with the constraints of the multi-neuron relaxation to prioritize branching.
- Cost Adjusted Branching, which aim to correct the score of the branching heuristic based on the expected computational cost induced by each possible branching choice.

Experiments are ran on MNIST and CIFAR10

[1] Beta-CROWN: Efficient Bound Propagation with Per-neuron Split Constraints for Complete and Incomplete Neural Network Verification
[2] PRIMA: Precise and General Neural Network Certification via Multi-Neuron Convex Relaxations

**Summary Of The Review:**

This paper combines existing techniques to improve the performance of complete verifiers. The contributions to the bounding process (incorporating the additional constraints with Lagrange multipliers) are relatively minor. The improvements to the branching procedure are interesting but should be clarified further.
It is however disappointing that the empirical evaluation has several weaknesses, mainly due to the lack of comparing to strong baselines.

---

> ### Author Response · Authors · 2021-11-17
> **Response to Reviewer 3m7j**
>
> $\newcommand{Rfi}{\textcolor{purple}{3m7j}}$
> We would like to thank reviewer $\Rfi$ for providing valuable feedback and raising interesting questions which we answer below.
>
> **How is the "expected bound improvement" for CAB computed?**
> We use the branching scores as a proxy for the “expected bound improvement”, as both BaBSR and FSB define their scores as the bound improvement under a local approximation and a cheaper bounding method, respectively, for an additional split. Indeed, this relationship becomes more qualitative when using active constraint scores. We have updated the formulation to make this clear.
>
> **Can you clarify why different branching decisions exhibit "considerable differences in computational cost”?**
> We do in fact mention at the end of section 3.2 that we keep intermediate bounds before the split layer fixed.
>
> When bounding an objective (e.g. a neuron's activation) with our method, only constraints on preceding neurons are considered. Hence, the intermediate bounds of neurons preceding the split are not tightened by the split constraint and hence not recomputed. This leads to a significant difference in the number of bounds to be computed. We added a detailed explanation to the appendix of how the split costs are computed to further clarify this point.
>
> **Can you compare MN-BaB to other methods on the ResNets in Table 2?**
> Yes, we do now compare to Beta-CROWN (where a public version supporting ResNets has been released recently) and ERAN/PRIMA. Please see Table 1 or the main response for a discussion.
>
> **Can you compare MN-BaB to other methods on a larger convolutional network?**
> Yes. We have added the larger ConvSuper, to Table 1 as suggested by $\Rf$. However, both OVAL and Beta-CROWN run out of memory on some samples on our hardware (RTX 2080 Ti), despite our best efforts to find settings reducing the memory requirements. Hence, we mostly focus our evaluation on the more relevant ResNets.
>
> **Why is ACS+CAB not used consistently as a branching method in Table 1?**
> Please see the main response. We now report ACS + CAB and BaBSR + CAB for all networks.
>
> **Why is verified accuracy and average runtime compared instead of cactus plots?**
> First, we believe that cactus plots are most suitable for a qualitative comparison of methods with very different performance characteristics, while becoming hard to interpret for similarly performing methods. Second, the robustness properties evaluated in Table 1 are not filtered to only include verifiable properties (in contrast to the often used, but mostly solved OVAL benchmark where cactus plots are typically used). In this setting, typically accuracy and average runtime is reported [1,2,3]. Third, while we agree that accuracy alone is not sufficiently informative, we believe that additionally comparing average runtime for a given (identical) timeout allows for an effective comparison between methods.
>
>
> **Why is eps=0.12 used for MNIST ConvSmall?**
> For this particular network, 0.12 is the perturbation size typically used in literature [1,2]. For the ConvSuper benchmark, we purposely deviated from the standard eps of 0.1 because even DeepPoly is sufficient to achieve 97% verified accuracy [2]. We choose an eps of 0.18 as a setting where DeepPoly was mostly unsuccessful but finding adversarial examples remains hard.
>
> We hope to have answered all of the reviewer’s questions and comments and look forward to the reviewer’s response to our rebuttal.
>
> **References**
> [1] Müller, Mark Niklas, et al. "PRIMA: Precise and General Neural Network Certification via Multi-Neuron Convex Relaxations." POPL2022
> [2] Singh, Gagandeep, et al. "Beyond the single neuron convex barrier for neural network certification." NeurIPS 2019.
> [3] Wang, Shiqi, et al. "Beta-crown: Efficient bound propagation with per-neuron split constraints for neural network robustness verification." ICML 2021 Workshop on Adversarial Machine Learning. 2021.

---

> > ### Comment · Reviewer_3m7j · 2021-11-22
> > **Thank you for clarifications.**
> >
> > I have read the authors response to my review, as well as to the other reviewers. I want to thank them for clarifying a lot of points in the process, and providing additional results.
> > Notably:
> > - Including additional networks in the evaluation of Table 1. It is a shame that due to memory issues, the other method based on mutli neuron (all inputs joint to one output of the ReLU in OVAL) could not be compared to the PRIMA style bounds (multiple inputs-outputs pair jointly). The comparison to Beta-Crown is already quite useful in showing the benefits.
> > - Adding FSB in the ablation study of Table 2
> > - Clarifying the branching section by explaining the expected bound improvement and the source of computational saving of CAB.
> >
> > This alleviates the problems that I had with the empirical evaluation, so I'm updating my score from a 5 to a 6. I'm not going higher because I still think that the novelty of the paper is relatively limited, although it's beneficial to have strong systems combining different existing ideas.
> >
> > One suggestion for a better exposition of results:
> > - If the contribution of the paper is on improving bound (and having better branching to get faster bound improvement), it might be beneficial to have some evaluation only on robust samples.
> > I think that aside from ERAN, none of the verifiers discussed here have any mechanism to look for violation to the spec subject to branching. As a result, any difference in speed in "vulnerable" samples is irrelevant to the contributions of the paper and it would make for clearer results to ignore them.

---

> > > ### Author Response · Authors · 2021-11-23
> > > **Reply to Reviewer 3m7j**
> > >
> > > We are happy that we were able to address the reviewer's questions and concerns.
> > >
> > > **Evaluated Properties**
> > > We followed the standard in the field of reporting results on all (first 100 or 1000) samples to avoid introducing any bias via the property selection process. Evaluating on samples known to be robust (e.g. union of all certified samples) would lead to the most accurate method (i.e. ours) being even faster as it would time out on the fewest samples (none where it is strictly more accurate). We thank the reviewer for their suggestion and are happy to include these results in the next revision of the paper.

---

### Official Review · Reviewer_gB2C · 2021-11-02

**Correctness:** 2
**Technical Novelty And Significance:** 3
**Empirical Novelty And Significance:** 3
**Recommendation:** 6
**Confidence:** 4

**Main Review:**

Strengths:

- Previously multi-neuron relaxations have shown to be instrumental in going beyond smaller networks [1][2]. Combining this in branch-and-bound framework is a novel idea that could potentially lead to interesting future work.
- Branching decisions are important in a BaB setting [5], the authors present interesting proposals (ACS + CAB) for both effective and efficient branching. Previous branching heuristics make branching decisions on expected bound improvements. Since MN-BaB operates on multi-neuron relaxations, branching heuristic ACS is presented that factors the improvements by multi-neuron relaxations into consideration.  CAB (cost-adjusted branching) postulates that runtime is a key factor in branching decisions for complete verifiers, hence proposing to do the branching decision as a function of expected bound improvement per cost instead of just expected bound improvement.
- The ablation studies show each system component is beneficial (Figure 3, 4, 5) and contributes to the overall improvements (Figure 6).

Weakness
- Experimental evaluation of the framework seems non-exhaustive and missing comparisons to important baselines (see more details below), making it hard to place MN-BaB (this paper) alongside the literature.

Reg. experimental evaluation:

Table 1: The authors claim that three out of the four networks are well-solved by previous methods and are less relevant, while for the fourth one, they perform the best. I have a few reservations:

- For the first three rows, it seems like MN-BaB isn’t the most effective and efficient method and for the fourth row the margin seems too small to claim “outperformance” -- hence more evaluation or analysis might be needed to better understand the behaviour.
- A possible remedy to this: Seems like there’s some promise in larger networks. Previous works have also considered ConvSuper [6], networks larger than ConvBig. Did the authors try to evaluate these benchmarks? What were the results?
- Another remedy for better understanding: Previous works [1][2][6][7][10][11][12] have all considered a larger set of benchmarks including more kinds of conv networks + FFNNs. I already mentioned conv networks above but the method presented seems to work for feed-forward networks (FFNNs) too but no evaluation is presented. Maybe because of space constraints and/or the argument that ERAN can already do very well. But it would be a good idea to have the results in the paper (or appendix) for a more holistic picture. While Beta-CROWN might not work for MLPs but comparisons to other baselines (including other variants of CROWN) would be useful.
- Reg. branching for MN-BaB: It’s unclear why three out of the four rows use BaBSR branching and the last one uses ACS. Was only the best performing branching heuristic reported? Since ACS is stated as one of the main contributions of the paper too, it would be good to see a breakdown of ACS vs. BaBSR for at least all these four benchmarks otherwise the MN-BaB results are not generated from a consistent set of options.
- Fixes needed (at minimum, the difficulty of fix in brackets):
            * ACS+CAB reporting for all benchmarks. (easy fix)
            * One larger conv network eval to claim “outperformance”. (medium fix)
- Fixes needed (ideally)
            * A couple of FFNNs evaluation for comparison with ERAN/OVAL. (larger change, can reuse numbers from PRIMA paper)


Table 2: The authors claim that their methods work on more challenging less regularized networks. This section is important and interesting and highlights how each component of the system contributes to the final results. My primary concern is that this experiment is in isolation and not placed well with the rest of the related works:
- For ResNet-6-A and ResNet-6-B: Was ERAN run since it can handle ResNets? What were the results? I agree OVAL does not support residual architectures. But other variants of CROWN that do support ResNet architectures and have a public implementation of the same [7][8][9], could have been evaluated (with alpha-beta-CROWN being the current best option).
- I raise this issue because the paper seems to claim “26% more samples while being around 30% faster” in multiple places but this seems to be on only one network (ResNet-6-A) against only one (weak) baseline for current SOTA (BaBSR). Comparing multiple benchmarks to a few relevant tools (at least one) would be a rigorous way to compute the performance improvements esp. since Table 1 is not shedding a lot of light.
- Fixes needed (at minimum, the difficulty of fix in brackets):
           * Report ERAN and a CROWN-variant (alpha-beta-CROWN is open-source, all options here [7][8][9]) in the table (medium change).
           * Fix the improvement claims to be relative to SOTA (best tool), averaged over all networks instead of just one. (easy change)

A suggestion regarding clarity: Section 2.3 briefly mentions two cases of unstable & stable ReLUs and different convex relaxations. This could be elaborated on some more, particularly on how it is incorporated and affects future decision choices.

Minor comments: Page 5: Enforcing split constraitns → Enforcing split constraints

Finally out of personal curiosity, it seems like the Beta-CROWN results in Table 1. were directly pulled from the paper [10]. Could the authors point me to the table/graph they referred to? Asking as I couldn’t find it myself.


- [1] PRIMA: Precise and General Neural Network Certification via Multi-Neuron Convex Relaxations. Müller et. al.
- [2] Beyond the Single Neuron Convex Barrier for Neural Network Certification. Singh et. al.
- [3] Branch and Bound for Piecewise Linear Neural Network Verification. Bunel et. al.
- [4] Scaling the Conve Barrier with Active Sets. De Palma et. al.
- [5] Improved Branch and Bound for Neural Network Verification via Lagrangian Decomposition. De Palma et.al.
- [6] An Abstract Domain for Certifying Neural Networks. Singh et.al.
- [7] alpha-beta-CROWN: https://github.com/huanzhang12/alpha-beta-CROWN
- [8] auto_LiRPA: https://github.com/KaidiXu/auto_LiRPA
- [9] CROWN-IBP: https://github.com/huanzhang12/CROWN-IBP
- [10] Beta-CROWN: Efficient Bound Propagation with Per-neuron Split Constraints for Complete and Incomplete Neural Network Verification. Wang et.al.
- [11] Automatic perturbation analysis for scalable certified robustness and beyond. Xu et.al.
- [12] Fast and Complete: Enabling Complete Neural Network Verification with Rapid and Massively Parallel Incomplete Verifiers. Xu et.al.


**Summary Of The Paper:**

The paper presents a novel framework for neural network verification by combining two popular paradigms in literature: tight multi-neuron relaxations [1][2] and branch-and-bound [3][4][5]. The benefit of such a framework is combing the best of both paradigms by improving the bounding method (multi-neuron relaxations) in a Branch-and-Bound framework resulting in scalability and completeness. Additionally, the authors also present a branching heuristic that can leverage their multi-neuron setup and propose a computation cost-adjusted branching to obtain decreased runtimes. They validate their hypothesis by comparing it to state-of-the-art verification tools and verifying established convolution network benchmarks for MNIST & CIFAR-10. They also present ablation studies for various features on two larger ResNet architectures showing how each system component contributes to improved results.


**Summary Of The Review:**

In conclusion, I feel the paper presents an interesting idea but needs a more thorough evaluation and comparisons to related works to justify the claims made. Hence,  the current submission is marginally below the acceptance threshold for ICLR.

---

> ### Author Response · Authors · 2021-11-17
> **Response to Reviewer gB2C**
>
> $\newcommand{Rf}{\textcolor{green}{gB2C}}$
>
> We would like to thank reviewer $\Rf$ for providing such detailed and valuable feedback, helpful comments and raising interesting questions which we answer below.
>
> **Can you extend the results in Table 1 to better support the claim of outperformance?**
> We did indeed only report the best performing branching heuristic for clarity. We have added an extra column and now always report ACS+CAB and BaBSR+CAB.
> We have added results on ConvSuper, where we outperform ERAN/PRIMA by a significant margin and were unable to find settings for Beta-CROWN and OVAL that did not run out of memory.
> Competitive performance on FFNs is typically only achieved when using extensive LP and MILP refinement of individual neuron bounds [1,2,3]. As we believe these networks to be of small practical relevance, we decided against optimizing for their verification. Instead, we now compare against Beta-CROWN and ERAN/PRIMA on our two (+ a new one) ResNets.
>
> **Can you compare MN-BAB better against related work on the challenging networks in Table 2?**
> We now compare against both ERAN/PRIMA and alpha-beta-CROWN on the two ResNets from Table 2 and the additional ResNet8-A. While all three tools perform similarly on the heavily regularized ResNet6-B, MN-BaB outperforms the other methods on the more challenging ResNet6-A and ResNet8-A.
> We have updated the wording to make it clear that the quoted improvements are maximal and not the average (against the best of the baselines). We believe this to be both the standard in the field and fair considering this improvement is obtained on the most challenging network analysed, which are the focus of this work.
>
> **Can you clarify the impact of a neuron being stable or unstable on their abstraction?**
> Yes, a ReLU neuron that is stable has been shown to either be always active (x>=0 => y=x) or inactive (x<=0 => y=0). Hence, their behaviour can be captured exactly with a linear function, which can be captured exactly by the DeepPoly domain and hence MN-BaB. Only unstable neurons, i.e. those which can not be shown to be always active or inactive, have to be approximated at all. The whole following discussion of the bounding process tackles the challenge of approximating these unstable neurons. We have clarified this.
>
> Thank you for pointing out the typo. We have fixed it. Please find the Beta-CROWN results in  Table 4 here https://openreview.net/pdf?id=Mm3gxxTfT7A
>
> We hope to have answered all of the reviewer’s questions and comments and look forward to the reviewer’s response to our rebuttal.
>
> **References**
>
> [1] Müller, Mark Niklas, et al. "PRIMA: Precise and General Neural Network Certification via Multi-Neuron Convex Relaxations." POPL2022
> [2] Singh, Gagandeep, et al. "Beyond the single neuron convex barrier for neural network certification." (2019).
> [3] https://github.com/huanzhang12/alpha-beta-CROWN

---

> > ### Comment · Reviewer_gB2C · 2021-11-21
> > **Thank you for the additional evaluations and your response**
> >
> > I would like to thank the authors for taking the time to run additional evaluations mentioned in my original review and addressing each of the several points raised. I have changed my score from 5 to 6 after reading through the latest version of the paper.
> >
> > **Can you extend the results in Table 1 to better support the claim of outperformance?**
> > Thank you for adding this table. Results on larger networks look better and the contrast between branching heuristics is good to see. I would recommend the authors complete evaluations to first 1000 images for the next version of the paper, for consistency. Best times are not bolded for rows CIFAR10 ConvSmall and CIFAR10 ResNet 8-A.
> >
> > **Can you compare MN-BAB better against related work on the challenging networks in Table 2?**
> > Table 1 answers this too. Thanks for adjusting the wording and fixing the percent improvement numbers. 28% improvement claim is now on the last network (CIFAR10 for ResNet8-A) where MN-BaB performs 23% while Beta-CROWN performs 18%.
> >
> > Thanks again!

---

> > > ### Author Response · Authors · 2021-11-21
> > > **Reply to Reviewer gB2C**
> > >
> > > We are happy that we were able to address all of the reviewer's concerns.
> > >
> > > **Clarification on highlighting best runtimes in Table1**
> > > We decided to only highlight the best runtime if the corresponding method also yields the highest verified accuracy as runtime and accuracy are tightly coupled for BaB based methods. Otherwise, in the most extreme case, a method that instantly returns “not verified” would always have the lowest runtime while adding no value.
> > >
> > > We will report results on the first 1000 samples for all networks for the next revision of the paper.

---

> > > > ### Comment · Reviewer_gB2C · 2021-11-21
> > > > **Response**
> > > >
> > > > Thank you for the clarifications!!

---

### Official Review · Reviewer_Ns4C · 2021-11-02

**Correctness:** 4
**Technical Novelty And Significance:** 2
**Empirical Novelty And Significance:** 2
**Recommendation:** 6
**Confidence:** 4

**Main Review:**

The method proposed  comprises (i) the integration of  multi-neuron relaxation
constraints  into a  bound propagation algorithm; (ii) the integration of ReLU
split constraints in said algorithm; (iii) the active constraint score and
cost-adjusted branching heuristics. Part (i) was previously developed and used
in a bound propagation-based tool (Eran). Part (ii) was previously developed
and used in a similar framework to the one in present paper (Wang et al.,
2021).  Therefore the novel contribution of the paper are the branching
heuristics.  These are however straightforward thereby (in my view) not
meriting an ICLR publication from a technical point of view.

Concerning the empirical evaluation, the paper carries out extensive ablation
studies whereby the effectiveness of the different constrains and heuristics
used is shown. It is definitely nice to see that these work in the context of
branch-and-bound. The comparison with related work is in my view weak however.
First, out of the four networks used, the present method marginally wins only
in two them in terms of average runtime and marginally wins only in one of them
in terms of verified accuracy. Whilst the authors claim that the efficacy of
their method is better observed in bigger networks, they do not include
comparisons with related tools on bigger networks. I think that the inclusion
of these in future revisions of the paper would significantly strengthen the
evaluation section.

Overall the paper is very well written; it was a joy to read. Some minor
comments:

- I did not find clear how the number of floating-point operations in the
  computation of the bounds is calculated in the cost-adjusted branching
  heuristic. In particular, how does this vary with different splits?

- Page 2. greater 0 -> greater than 0.
kkk

**Summary Of The Paper:**

The paper puts forward a method for the formal verification of ReLU-based
neural networks. The method incorporates  previously developed multi-neuron
relaxation and split constraints in a bound propagation algorithm which is used
in a branch-and-bound procedure. Experimental results are reported that compare
the procedure with related methods on a number of  MNIST and CIFAR10 models.
Ablation studies are also carried out on ResNet6-A and ResNet6-B.

**Summary Of The Review:**

Whilst the paper introduces some novel branching heuristics for the complete
verification of ReLU neural networks via branch-and-bound, the overall
contribution lacks in my opinion the sufficient novelty to merit an ICLR
publication. I think that  the authors should include additional empirical
comparisons with related tools on bigger networks and consider submitting a
tool paper in a top venue.

---

> ### Author Response · Authors · 2021-11-17
> **Response to Reviewer Ns4C**
>
> $\newcommand{Rth}{\textcolor{red}{Ns4C}}$
>
> We would like to thank reviewer $\Rth$ for providing valuable feedback and raising interesting questions which we answer below.
>
> **Were PRIMA multi-neuron constraints already integrated into a bound-propagation-based algorithm as part of ERAN?**
> No, PRIMA uses the multi-neuron constraints to tighten an LP encoding that is then solved using the CPU-based solver GUROBI. This constitutes the main bottleneck which is only alleviated by the efficient use of partial MILP encodings.  For more details, we refer to section 6 of Müller et al. [1]. This work, in contrast, leverages multi-neuron constraints as part of the bound propagation enabling an efficient GPU implementation. This is in fact the largest contribution of this work.
>
> **Can you provide a more thorough comparison to related work?**
> Yes. We do now compare to both ERAN and Beta-CROWN on ResNet6-A and -B, ConvSuper and a new ResNet8-A. We observe that MN-BaB consistently outperforms these methods (with Beta-CROWN running out of memory on ConvSuper) on the challenging networks while performing similar on the heavily regularized ResNet6-B. Please see Table 1 for more details.
>
> **How is the number of floating point operations determined for cost adjusted branching?**
> We approximate the number of floating point operations as follows: Bounding an objective via backsubstitution is dominated by a series of matrix multiplications. Since the number of neurons and prima constraints, which determine the sizes of these matrices, are known, we can compute the cost of these multiplications. The total cost of a split is then the sum of the backsubstitution-cost of all the intermediate bounds that need to be recomputed. We have added a more detailed description to Appendix X.
>
> Thank you for pointing out the Typo. We have corrected it.
>
> We hope to have clarified the novelty of our work, answered all of the reviewer’s questions and would like to ask the reviewer to consider these points when updating their review in response to this rebuttal.
>
> **References**
> [1] Müller, Mark Niklas, et al. "PRIMA: Precise and General Neural Network Certification via Multi-Neuron Convex Relaxations." POPL2022

---

> > ### Comment · Reviewer_Ns4C · 2021-11-22
> > **Thank you for your response**
> >
> > I thank the authors for clarifying the support of Eran of multi-neuron
> > constraints, the computation of floating point operations, and for providing
> > additional experimental results. On the one hand I still think that the
> > technical contribution of the paper is very incremental to previous work, but
> > on the other hand the additional experimental results strengthen significantly
> > the empirical evaluation of the method. Given its superior performance w.r.t
> > state-of-the-art, I think that the method is worth disseminating to the scientific
> > community. Given this I increased my score from 5 to 6.

---

### Official Review · Reviewer_qGFq · 2021-11-03

**Correctness:** 4
**Technical Novelty And Significance:** 3
**Empirical Novelty And Significance:** 3
**Recommendation:** 6
**Confidence:** 3

**Main Review:**

The paper is clearly written, with a clean, linear presentation of the method and its derivation. The "novelty" of the work is somewhat incremental, but the new methods show their efficacy, and the paper is a worthy contribution to the literature.

* I wish that the authors spent more time describing the work of Muller et al that they build on for the multi-neuron constraint generation. Consider adding a brief discussion.
* p2: To what extent is your framework "applicable" to network with Sigmoid and Tanh activations? If it involves approximating the nonlinear activations, is it still sound and exact?
* p2: I think you need to add a "sound" to "A method is called complete if..."?
* p3: "we derive linear upper and lower bounds of the form z^i >= Az^(i-1) + c for each...". This seems like it's only lower bounds?
* p6: Is "noisy proxies" the right choice of words?
* p8: Is there a fundamental reason beta-Crown and Oval don't support ResNet, or is it simply a matter of implementation? What about ERAN?
* The authors mention a "GPU-based dual optimizer" in the abstract and introduction, but there is no explicit mention of GPU compatibility of the method in the body of the paper. Can the authors clarify?

**Summary Of The Paper:**

This paper presents a branch-and-bound algorithm for neural network verification that uses a dualized variant of an existing approach for generating multi-neuron relaxations. Computational results are presented showcasing the quality of the verification bounds, and the scalability of the method.

**Summary Of The Review:**

The paper presents its new idea cleanly, and the improvements appear substantial enough for publication.

---

> ### Author Response · Authors · 2021-11-17
> **Response Reviewer qGFq**
>
> $\newcommand{Rt}{\textcolor{blue}{qGFq}}$
>
> We would like to thank reviewer $\Rt$ for providing valuable feedback and raising interesting questions which we answer below.
>
> **Could you describe the work of Muller et al [1] on multi-neuron constraint generation in more detail?**
> Since our method is general enough to be combined with arbitrary (multi-neuron) constraints, we decided not to discuss the (complex) specifics of PRIMA constraint computations in this work. Below, we give a short summary and refer the interested reader directly to [1].
> The PRIMA Multi-Neuron constraints [1] are based on the same idea as those introduced in Singh et al. 2019 [2], but use approximate algorithms for a speed-up of multiple orders of magnitude. At a high level, the neurons of an activation layer are first grouped into sets of size 3 or 4 before an octahedral approximation of their input is computed. Then an (approximate) convex hull of the (6d or 8d) input output space of these neurons is computed, which can be represented by a set of (Multi-Neuron) constraints. By intersecting many of these constraint sets, complex interactions can be captured.
>
> **To what extent is your framework "applicable" to network with Sigmoid and Tanh activations?**
> Both the single neuron relaxations [3], as well as the multi-neuron constraints [1] are also applicable to sigmoid and tanh activation functions. Thus the bounding method underlying MN-BaB is directly applicable without changes. Branching can also be performed in the same manner as for ReLUs. The main difference is that sigmoidal functions cannot be captured exactly with linear constraints even after branching. Therefore, the resulting Branch-and-Bound verifier will not be complete.
>
> **Did you mean to add a "sound" to "A method is called complete if..."?**
> In line with early work on NN verification [4], we use the following definition of completeness: A method is called complete if it certifies any property that actually holds. This does not necessarily imply soundness, the guarantee that every certified property actually holds. We call a method that is both sound and complete, exact. However, since all NN verification approaches are required to be sound, the term “complete” is often (and also by us) used instead of “exact”.
>
> **It seems like you only describe lower bounds at the top of page 4(?)**
> Indeed, these are only the lower bounds. We derive the corresponding upper bounds analougosly by lower bounding the negative value -z^(I).
>
> **Is "noisy proxies" the right choice of words?**
> We believe “proxy” to be appropriate since FSB and BaBSR both approximate the bound improvement obtained via splitting when bounding with DeepPoly style bounds. However, neither uses the actual bounding method to compute these improvements. The larger the (qualitative) difference between the actually applied bounding method and the DeepPoly/CROWN style bounds underlying the approximation, the less predictive the obtained branching score for the actual bound improvement. This difference is much bigger to a method using  multi-neuron constraints than to one only optimizing the slopes of the same transformer. We have updated our writing to clarify this.
>
> **Is there a fundamental reason Beta-CROWN and Oval don't support ResNet, or is it simply a matter of implementation?**
> To the best of our knowledge, there is no fundamental reason why the frameworks  Oval, Beta-CROWN or ERAN would not support ResNets. ERAN does in fact support them and Beta-CROWN has since original submission been updated to a new version (alpha-beta-CROWN) which also supports ResNets. We have now evaluated both ERAN and Beta-CROWN on all three ResNet benchmarks and observe that MN-BaB outperforms both other methods on the less regularized ResNet6-A and ResNet8-A while performing similarly on the heavily regularized ResNet6-B. See Table 1 for more details.
>
> **Can the authors clarify the GPU compatibility of their tool?**
> The bounding procedure described in section 3.1 yields an optimization problem in the lagrange parameters alpha, beta and gamma. This objective can be optimized using a GPU-based autograd framework. We use a Pytorch implementation as described in Section 4. We have made this more clear in the updated paper.
>
> We hope to have answered all of the reviewer’s questions, are happy to provide more detail and look forward to the reviewer’s response to our rebuttal.
>
> **References**
> [1] Müller, Mark Niklas, et al. "PRIMA: Precise and General Neural Network Certification via Multi-Neuron Convex Relaxations." POPL2022
> [2] Singh, Gagandeep, et al. "Beyond the single neuron convex barrier for neural network certification." NeurIPS 2019
> [3] Singh, Gagandeep, et al. "An abstract domain for certifying neural networks." POPL 2019
> [4] Katz, Guy, et al. "Reluplex: An efficient SMT solver for verifying deep neural networks." ICCAV 2017

---

### Official Review · Reviewer_WZm2 · 2021-11-03

**Correctness:** 2
**Technical Novelty And Significance:** 3
**Empirical Novelty And Significance:** 3
**Recommendation:** 6
**Confidence:** 4

**Main Review:**

In general I like the directions that this paper is taking to push the state-of-the-art of neural network verification. The contributions are novel to my knowledge. The encoding of the verification problem using Lagrange multipliers looks sound. The two ideas on branching heuristics are also interesting. After reading the previous sections, I expect the experimental evaluation to demonstrate two points:
1) the GPU-based dual algorithm is better than an (almost equivalent) MILP-encoding;
2) ACS and CAB are competitive with SOTA branching heuristics.

The experiment on ResNets suggest does seem to show that CAB is beneficial for both BABSR and ACS.  However, I'm not fully convinced of the efficacy of the rest of the proposed techniques. My concerns and questions are listed below:
1. In Table 1, why does MB-BAB use different branching heuristics for different networks? Maybe show the numbers of BABSR+CAB and ACS+CAB in two different columns?
2. Table 1 suggests that ERAN (PRIMA + MILP-encoding) solves the most instances overall, while being slower on only 1 network. To draw the conclusion that Gurobi (with 16 cpus) is less scalable than a GPU-based procedure on the verification problem (which is the motivation of the proposed encoding of the verification problem), more experiments are needed than a single perturbation bound and a single network.
3. In Table 2, why is "DeepPoly only" instead of "PRIMA only" (or kReLU only) used as a filter for easy verification problem.
4. What are the number of ReLUs in the ResNet architecture and in the ConvBig/Small networks? I saw the concrete architectures in the Appendix but since the motivation for excluding ERAN from Table 2 is the claim that MILP-encoding is less scalable than the GPU-based approach on larger models, the architecture information (or at least the number of ReLUs) seems to be relevant enough to be put in the main paper.
5. There are more recent branching heuristics than BABSR as mentioned in the related work. It would strengthen the paper to use one of those (e.g., FSB) as the baseline.

Overall, while I find the proposed techniques promising, a more thorough evaluation is needed to show their benefits.

Minor comments:
- Typo: page 5: Enforcing split constraitns
- Figures 5, 6 are compressed in a way that favors the proposed techniques. Maybe present it with the same width and height?

**Summary Of The Paper:**

The paper introduces a complete verification procedure that 1) encodes PRIMA constraints as a Lagrangian function, which admits GPU-based algorithm; 2) uses the values of the Lagrange parameter to guide the branching (ACS); and 3) regularizes the branching score with the estimated cost for a split.

**Summary Of The Review:**

Promising verification procedure, but a more thorough evaluation is needed.

---

> ### Author Response · Authors · 2021-11-17
> **Response to Reviewer WZm2**
>
> $\newcommand{Ro}{\textcolor{orange}{WZm2}}$
>
> We would like to thank reviewer $\Ro$ for providing valuable feedback and raising interesting questions which we answer below.
>
> **Why are results for MN-BaB reported with different branching heuristics in Table 1?**
> Please see the main response. We now report ACS + CAB and BaBSR + CAB for all networks.
>
>
> **Can you provide more results showing that the CPU based ERAN is less scalable than the proposed GPU-based MN-BaB?**
> First, we want to point out that ERAN, as used here, is not fully CPU based, but uses GPUPoly to compute all initial bounds including the inputs required for the PRIMA constraints. Switching to a fully CPU based version empirically increases verification times by more than a factor two.
> Nevertheless, we are happy to provide more results and now additionally compare on ConvSuper [1], the new ResNet8-A and both ResNet6-A and B, where we observe MN-BaB significantly outperforming ERAN and other state-of-the-art-methods on the challenging networks (see Table 1).
>
>
> **Why is "DeepPoly only" used as a baseline in Table 2 instead of "PRIMA only"?**
> There are two reasons for this. First, we designed Table 2 to be a linear ablation study from the most simple subset of our approach (DeepPoly) to the full MNBaB ACS+CAB in order to analyze the improvement of the contributions separately. Second, “PRIMA” typically uses a significant amount of MILP refinement making it even harder to place it fairly in an ablation study. We have now added DeepPoly + Multi-Neuron Constraints (no branching) to the ablation study.
>
> **What are the number of ReLUs in the ResNet architecture and in the ConvBig/Small networks?**
> We would like to clarify that by scalable we do not necessarily mean larger models but less regularized. If sufficiently regularized, even ResNet34s become reasonably easy to verify but lose most of their performance (retaining e.g. only 35% natural accuracy) in the process [2]. Nevertheless, we have included the number of ReLUs in appendix A.
>
> **Why are no more recent branching heuristics than BaBSR (e.g., FSB) used as the baseline?**
> Most of the more recent branching heuristics e.g. FSB [3] or DeepSplit [4] use more expensive methods to more accurately predict the bound improvement obtained after splitting and bounding using single-neuron constraints. We believe and confirm empirically that these methods do not predict well the bound improvement well in a multi-neuron setting. In combination with their increased computational cost, this makes them unattractive for use in MN-BaB. We have now added them to our ablation study in Table 2.
>
> **Are figures 5 and 6 compressed in a way that favors the proposed techniques?**
> We made sure to use the same ranges for both axes and believe that combined with the inclusion of the dashed equal-time line, this allows for a fair comparison. However, we have now updated both plots to ensure that the axes are scaled equally.
>
> We hope to have answered all of the reviewer’s questions, are happy to provide more detail and look forward to the reviewer’s response to our rebuttal.
>
> **References**
> [1] Singh, Gagandeep, et al. "An abstract domain for certifying neural networks." Proceedings of the ACM on Programming Languages 3.POPL (2019): 1-30.
> [2] Müller, Christoph, et al. "Neural network robustness verification on gpus. CoRR abs/2007.10868 (2020)." (2007).
> [3] De Palma, Alessandro, et al. "Improved Branch and Bound for Neural Network Verification via Lagrangian Decomposition." arXiv preprint arXiv:2104.06718 (2021).
> [4] Henriksen, Patrick, and Alessio Lomuscio. "DEEPSPLIT: An efficient splitting method for neural network verification via indirect effect analysis." Proceedings of the 30th international joint conference on artificial intelligence (IJCAI21). To Appear. ijcai. org. 2021.

---

> > ### Comment · Reviewer_WZm2 · 2021-11-17
> > **Thank you for your response**
> >
> > Thanks for providing additional results. I have raised my score to 5 since the authors have addressed all my questions, which allows me to re-evaluate the paper.
> >
> > I agree Table 1 does show that the CPU based ERAN performs worse than the proposed GPU-based MN-BaB (in particular on ResNet-8). I'm curious whether the MILP solver would perform better without PRIMA constraints (but just take the tightened bounds and use a single neuron encoding for instance). But I am in general convinced of the benefit of a GPU encoding.
> >
> >  On the other hand, the augmented Table 2 seems to show that a majority of the problem does not actually require any search (e.g., ACS +CAB solved only 6 more instances than PRIMA alone for ResNet 6-b). Given that complete search (in particular the branching heuristics) is a focus of this paper, it seems better to perform a more careful benchmarking (maybe try different perturbation bounds) to include more benchmarks that cannot be solved without branching. This would make the evaluation much stronger.

---

> > > ### Author Response · Authors · 2021-11-18
> > > **Reply to Reviewer WZm2**
> > >
> > > We are happy that we were able to address the reviewer's main concern.
> > >
> > > **MILP Encodings**
> > > In fact, the PRIMA constraints are only used for partial MILP encodings (starting from an LP encoding, some neurons in the final layers are encoded using binary variables to increase precision efficiently). They are dropped for full MILP encodings, just as the reviewer suggested.
> > >
> > > **Challenging Benchmarks**
> > > We agree that heavily regularized networks and easier benchmarks such as ResNet6-B are not suitable to benchmark the next generation of neural network verifiers. This is actually exactly the point we wanted to highlight by the comparison to the much less regularized ResNet6-A of the same architecture and the reason why we also evaluate on the more challenging ConvSuper, ResNet8-A, and ResNet6-A. On these networks, the full MN-BaB certifies 2-times, 56%, and 70% more samples than a version with no branching, respectively, while also outperforming the state-of-the-art verifiers ERAN/PRIMA and Beta-CROWN. We think that this quite convincingly demonstrates that, especially on the challenging networks that the reviewer asked us to consider, MN-BaB performs significantly better than other state-of-the-art methods.

---

> > > > ### Comment · Reviewer_WZm2 · 2021-11-22
> > > > **Thank you for your response**
> > > >
> > > > I would like to thank the authors for the clarification. I believe after the revision the experimental section looks quite convincing. So I raised my score from 5 to 6.

---

### Author Response · Authors · 2021-11-17
**General Response**

$\newcommand{Ro}{\textcolor{orange}{WZm2}}$
$\newcommand{Rt}{\textcolor{blue}{qGFq}}$
$\newcommand{Rth}{\textcolor{red}{Ns4C}}$
$\newcommand{Rf}{\textcolor{green}{gB2C}}$
$\newcommand{Rfi}{\textcolor{purple}{3m7j}}$


We would like to thank all reviewers for their helpful feedback, interesting comments, and insightful questions. We are happy to hear that the reviewers found our work on efficiently leveraging multi-neuron constraints in a Branch-and-Bound framework to be an interesting direction ($\Ro$, $\Rf$) and a worthy contribution to the literature ($\Rt$), and we appreciate the positive reception of our branching heuristics ($\Ro$, $\Rf$, $\Rfi$). We identified two main reasons why reviewers did not recommend our work for acceptance and addressed them below, before answering additional questions in more detail in individual responses to every reviewer.

**Establish that MN-BaB outperforms other state-of-the-art tools in a more thorough comparison  ($\Ro$, $\Rth$, $\Rf$, $\Rfi$)**
We have significantly extended the comparison of MN-BaB with other state-of-the-art tools, especially in challenging settings. Namely, we have added two completely new networks ConvSuper from [1] and ResNet8-A and now also compare to ERAN and Beta-CROWN on both of these as well as ResNet6-A and ResNet6-B.  We are happy to report that these results confirm the previously observed trends, with MN-BaB performing best on the challenging networks and all three tools performing similarly on the heavily regularized ResNet6-B. Please see the updated Table 1.

**A more thorough comparison of ACS and BaBSR (+CAB) ($\Ro$, $\Rf$, $\Rfi$)**
We have now updated Table 1 to  include results for both ACS + CAB and BaBSR + CAB on all 8 networks we consider and added a discussion of the observed performance characteristics. At a high level, ACS leverages information from enforcing the multi-neuron constraints. The more interactions between same-layer neurons a network has, the more multi-neuron constraints will be active and tighten the optimization problem and consequently the more information is captured in the active constraint scores. The less challenging networks exhibit relatively few of these complex interactions hence leading to a weaker performance of ACS.

We have found two typos when updating the results, namely the mean runtime of OVAL on CIFAR10 ConvSmall was actually 17.7s instead of 7.2s and the ablation study in Table 2 was conducted at epsilon 1/255 and not 2/255. We have corrected them and all typos pointed out by the reviewers in the updated version.

We hope to have removed any remaining doubts regarding the performance of MN-BaB with this significant extension of our experimental evaluation and are looking forward to the reviewer’s responses to our rebuttal.

**References**
[1] Singh, Gagandeep, et al. "An abstract domain for certifying neural networks." Proceedings of the ACM on Programming Languages 3.POPL (2019): 1-30.

---

### Decision · Program_Chairs · 2022-01-20

**Decision:**

Accept (Poster)

**Comment:**

The authors improve upon existing algorithms for complete neural network verification by combining recent advances in bounding algorithms (better bounding algorithms under branching constraints and relaxations involving multiple neurons) and developing novel branching heuristics. They show the efficacy of their method on a number of rigorous experiments, outperforming SOTA solvers for neural network verification on several benchmark datasets.

All reviewers agree that the paper makes valuable contributions and minor concerns were addressed adequately during the rebuttal phase. Hence I recommend that the paper be accepted.